# Field-Based Gait Retraining to Reduce Impact Loading Using Tibial Accelerometers in High-Impact Recreational Runners: A Feasibility Study

**DOI:** 10.3390/s25061712

**Published:** 2025-03-10

**Authors:** Eoin W. Doyle, Tim L. A. Doyle, Jason Bonacci, Joel T. Fuller

**Affiliations:** 1Faculty of Medicine, Health, and Human Sciences, Macquarie University, Sydney, NSW 2113, Australia; 2Biomechanics, Physical Performance, and Exercise Research Group, Macquarie University, Sydney, NSW 2113, Australia; 3School of Exercise and Nutrition Sciences, Deakin University, Melbourne, VIC 3125, Australia

**Keywords:** biomechanics, injury prevention, impact, kinematics, kinetics, running

## Abstract

This study investigated the feasibility of a field-based gait retraining program using real-time axial peak tibial acceleration (PTA) feedback in high-impact recreational runners and explored the effects on running biomechanics and economy. We recruited eight recreational runners with high landing impacts to undertake eight field-based sessions with real-time axial PTA feedback. Feasibility outcomes were assessed through program retention rates, retraining session adherence, and perceived difficulty of the gait retraining program. Adverse events and pain outcomes were also recorded. Running biomechanics were assessed during field and laboratory testing at baseline, following retraining, and one-month post-retraining. Running economy was evaluated during laboratory testing sessions. Seven participants completed the retraining program, with one participant withdrawing due to illness before commencing retraining. An additional participant withdrew due to a foot injury after retraining. Adherence to retraining sessions was 100%. The mean (SD) perceived difficulty of the program was 4.3/10 (2.2). Following retraining, the mean axial PTA decreased in field (−29%) and laboratory (−33%) testing. The mean instantaneous vertical loading rate (IVLR) reduced by 36% post-retraining. At one-month follow-up, the mean axial PTA remained lower for field (−24%) and laboratory (−34%) testing, and the IVLR remained 36% lower than baseline measures. Submaximal oxygen consumption increased following gait retraining (+5.6%) but reverted to baseline at one month. This feasibility study supports the use of field-based gait retraining to reduce axial PTA and vertical loading rates in recreational runners without adversely affecting the running economy.

## 1. Introduction

Running is a popular form of exercise with high global participation rates [1]. Despite its popularity, running is associated with a high injury risk, especially in novice and recreational runners [2,3]. Running-related injuries have a multi-factorial aetiology [4] and are often related to the overloading of the musculoskeletal system due to repeated microtrauma [5]. The prevalence of running-related injuries has prompted researchers to explore new methods for preventing these injuries.

Modifiable risk factors, such as running biomechanics, have been investigated to help inform injury prevention strategies with varied results [6,7,8]. Impact loading variables have been identified as potentially relevant modifiable risk factors [9]. Elevated vertical loading rates have been linked to the development of running-related injuries in prospective [10,11,12] and retrospective [9,13,14] studies. In contrast, others have found no association between the loading rate and subsequent running-related injuries [15]. It has been proposed that potential running-related biomechanical risk factors, such as vertical loading rates, may be injury-specific [7]. Several retrospective studies have identified higher vertical loading rates in runners with specific injuries, including plantar fasciitis [9], tibial bone stress injuries, [10,14,16,17] and patellofemoral pain [9,18,19], compared to uninjured runners.

Altering one’s running technique through gait retraining is a promising strategy to reduce the risk of running-related injuries in runners [20,21]. Data from two randomised controlled trials (RCT) that included impact-based gait retraining demonstrated decreased injury risk (hazard ratio 0.38 [95% confidence interval, 0.25 to 0.59]) [22] and a rate ratio of 0.52 [95% CI, 0.31–0.86] [23] in retrained runners compared to the controls at one-year follow-up. These interventions consisted of laboratory-based retraining sessions with real-time visual feedback and verbal cueing to target a reduction in vertical loading rates. However, one of the challenges is transferring these strategies from laboratory environments into ecologically valid field settings. Expensive biomechanics laboratories are not readily available to most clinicians and runners, so the broad relevance and impact of existing laboratory approaches are notably reduced.

Using inertial measurement units (IMU), runners can receive real-time feedback on specific measures in field settings, such as tibial acceleration. Notably, the axial component of peak tibial acceleration (PTA) has been closely associated with laboratory measures of vertical loading rates, with moderate to strong correlations observed in both healthy (r = 0.64–0.97) [24,25,26] and injured runners (r = 0.66–0.82) [27]. Therefore, providing PTA feedback to runners allows for a field-based solution that may elicit similar benefits to laboratory studies that targeted a reduction in vertical loading rate and reported significant injury reductions [22].

Most studies that have provided feedback on axial PTA occurred in a laboratory setting [28,29,30,31]. Recently, researchers have used real-time feedback of axial PTA in field settings [32,33]. Morris et al. [32] conducted an RCT that provided runners with real-time biofeedback using a mobile system during their usual training. The authors attempted to reduce axial PTA below 6g, as this was the value suggested whereby runners would adopt a non-rearfoot footstrike. Despite reporting compliance issues with the sensor, the authors demonstrated a reduced risk of knee pain at one-year follow-up in the retraining group (relative risk [RR] 5.6 [95% CI 1.9 to 16.8]), albeit with a higher risk of foot pain (RR 2.6 [95% CI 0.8 to 8.4]). More recently, Van den Berghe et al. [33] conducted a quasi-RCT where runners with high axial PTA were provided with music-based biofeedback during six supervised running sessions of 20 min over 3 weeks through a body-worn system to reduce axial PTA by 30% of baseline levels. Following the retraining program, the authors reported significant reductions (26%) in axial PTA in runners who received feedback.

Furthermore, acute changes in running technique have been shown to increase cognitive demand [34] and impair running economy in the short term [35]. However, studies with a one-month follow-up post-retraining have not found differences in running economy [31,36], which indicates that economy changes may return towards baseline levels over time. It remains important to consider whether providing axial PTA feedback may lead to adverse outcomes, such as injury and pain, or short-term reductions in running economy.

There is a lack of research exploring field-based gait retraining using real-time axial PTA feedback. Additionally, previous studies using PTA feedback have required supervised retraining [33] or encountered issues with sensor compliance [32]. Examining the feasibility of field-based retraining using PTA feedback is an important step in exploring the suitability of this approach and identifying any potential challenges before scaling up to larger efficacy trials. Therefore, the primary aim of this study was to investigate the feasibility of a field-based gait retraining program using real-time axial PTA feedback in healthy recreational runners with high landing impacts. A secondary aim was to examine the time course changes of the program on running biomechanics and economy.

## 2. Materials and Methods

This study used a single-arm trial design to examine the feasibility of gait retraining using real-time axial PTA feedback during field sessions in healthy recreational runners with high landing impacts. Biomechanical and running economy outcomes were assessed following field-based gait retraining and after one month of returning to usual training.

### 2.1. Participant Eligibility and Recruitment

Recreational runners were recruited from existing participant databases associated with previous running biomechanics studies undertaken at Macquarie University, community running clubs, and social media adverts. Interested runners were screened using a REDCap online survey [37,38]. Participants were eligible if they were aged 18–50 years, owned or had access to an iOS smartphone (necessary for recording field-based outcomes through an iOS application), and were fluent in English (to follow retraining instructions). Additionally, only participants who self-reported running at least 10 km per week (on average) over the previous three months and with a reported ability to run 5 km in under 26 min were eligible for inclusion (to minimise the risk of injuries due to a lack of familiarity with regular running [39]). Runners were excluded if they presented with a recent injury within the previous six weeks or reported a history of lower limb surgery or any neurological, inflammatory, or rheumatoid disease that affected their running gait. All participants provided written informed consent before enrolling in the study. Ethics approval was granted by the Macquarie University Ethics Committee (Reference No: 520211086135306). A total of eleven participants (eight males and three females) were recruited to participate in the study.

### 2.2. High-Impact Screening

Eligible runners were invited to an initial screening test to identify their typical landing impacts during running. Only high-impact runners with an axial PTA > 9 g while running at 3.7 m/s were eligible to participate in the study. An axial PTA threshold of 9 g was chosen based on a prospective report by Davis et al. [40], indicating that runners with a PTA above 9 g are at an increased risk of sustaining a tibial stress fracture. This criterion has been used in numerous other retraining studies [31,33,41]. PTA was assessed using two wireless IMUs (Blue Trident, iMeasureU, Auckland, New Zealand) that measure 42 × 27 × 11 mm with a mass of 9.5 g. The IMUs were attached to the left and right distal medial tibia (1 cm above the superior border of the medial malleolus) using the manufacturer’s strap to ensure IMU alignment with the tibia’s vertical axis (Figure 1). IMUs were secured with elastic cohesive tape (Victor, Whiteley Medical Supplies Pty Ltd., Sydney, NSW, Australia). Participants wore a single pair of their usual running shoes for baseline testing and then used the same shoes at all subsequent testing. This controlled for any effects of changing footwear between tests. Following a warm-up consisting of five minutes of walking or running and self-selected dynamic stretching, the axial PTA of both limbs were assessed during a 3-min run on an instrumented treadmill (AMTI Compact Tandem, Watertown, MA, USA) at 3.7 m/s. For screening, using an iOS application, PTA was sampled at 562 Hz (real-time insight) over 60 s during the final minute of running. Runners who exceeded impact thresholds on either limb were classified as high-impact runners and considered eligible for the study intervention. Runners were unaware of the timing of axial PTA sampling and were blinded to the inclusion criterion of high-impact landings for the duration of their participation.

### 2.3. Laboratory Baseline Testing

Within two weeks of screening, participants identified as high-impact runners attended laboratory testing (baseline). Participants were instructed to wear their usual running shoes for all data collection testing. Participants were fitted with IMUs on each lower limb as well as a University of Western Australia (UWA) full-body reflective marker set [42]. Kinematic data were collected at 250 Hz by an eight-camera motion capture system (Vicon, Oxford, UK). Simultaneously, force and acceleration data were collected from the instrumented treadmill and IMUs at 1000 and 1125 Hz, respectively.

Following a 5-min warm-up, participants performed two 4-min runs on the treadmill at their self-selected preferred speed. Once participants had selected their preferred running speed, that same speed was used for all subsequent testing to control for any effects of changing speed. Participants were instructed to choose a pace they were “*comfortable maintaining for at least 30 min of continuous running*” and equivalent to a 12/20 rating using the Borg Rating of Perceived Exertion (RPE) [43]. Motion capture assessment was completed during the first run. For the second run, reflective markers were removed, and participants were fitted with a portable metabolic analysis system (K5, COSMED, Rome, Italy) to capture sub-maximal oxygen consumption. Heart rate (HRM-Dual^®^, Garmin, Olathe, KS, USA) and perceived exertion (Borg RPE) were recorded.

### 2.4. Gait Retraining Intervention

Following baseline testing, participants received instructions on using the IMU as a feedback device during a short treadmill run at their preferred speed. During this run, participants received audible feedback through an application (CaptureU, iMeasureU, Auckland, New Zealand) on an iOS tablet (iPad Air 2, Apple, Cupertino, CA, USA), with the display mirrored to a monitor (Figure 2). A threshold limit of 80% of participants’ axial PTA recorded during field testing was used as the retraining target (20% reduction from baseline measures) and displayed as a shaded target area on an external monitor. Participants could view their axial PTA peaks in real time and were instructed to keep their acceleration peaks within the shaded area. If runners exceeded their individualised axial PTA threshold on either limb, they were alerted by a loud audible sound. If participants were unable to reduce their axial PTA below the targeted threshold, they were provided with verbal cues [20] to “*increase your step rate*”, “*reduce your heel strike*”, or “*soften your landing noise*”.

Participants were instructed to complete eight field-based overground runs with real-time feedback progressing from 15 to 30 min over the subsequent 2–3 weeks (Figure 3) [28]. Participants were asked to complete these runs on a flat or mostly flat running surface or oval with a firm surface (asphalt or hard-packed trail). During the retraining period, participants were instructed to refrain from other running training apart from warm-ups and cool-downs for each session. Participants were provided pre-programmed sessions (Figure 3) on a GPS watch (Garmin Instinct^®^, Garmin Ltd., Olathe, KS, USA). During each run, participants wore an IMU on each lower limb at the distal tibia following illustrated instructions to align the IMUs using the manufacturer’s strap and securely fasten them using cohesive tape (Table A1). Real-time axial PTA feedback was provided through the paired smartphone application, whereby participants received audible feedback when they exceeded their target axial PTA threshold on either limb. The audible feedback pitch differed between limbs, so participants could discern which limb exceeded the target axial PTA threshold. Participants only received audible feedback throughout their run, as using visual feedback on their smartphone application while running was impractical. As per the laboratory session, a threshold of 80% of participants’ screening axial PTA was selected as the feedback threshold. A faded feedback program was used to facilitate motor learning [44] and the internalisation of the new gait pattern [29]. Audible feedback was gradually removed during the last four runs as per previous approaches (Figure 3) [28,29,45]. To enable the faded feedback, haptic and audible notifications were provided to participants using the Garmin GPS watch to either use or not use feedback at pre-programmed intervals. Participants could remove feedback through the volume control button on their headphones. Participants were provided with feedback on running speed through their GPS watches and instructed to keep within the pre-defined pacing zones (±5%) of their preferred running speed.

#### 2.4.1. Laboratory Follow-Up and Retention Testing

After completing eight field-based runs, participants returned to the laboratory to have their running biomechanics and sub-maximal running economy re-assessed (follow-up). Assessment procedures from the baseline were repeated. Following this, participants resumed their regular training for one month before returning to the laboratory for final testing (retention).

#### 2.4.2. Field-Based Testing

Participants’ field-based running biomechanics and heart rate were assessed during a standardised 2.7 km outdoor run (a mostly flat 480 m oval concrete track), performed at baseline, follow-up, and retention time points. All field testing was performed on the same outdoor track and controlled for any effects of changing surfaces between tests. For these runs, participants were provided with a pre-programmed GPS watch (Instinct^®^, Garmin, Olathe, KS, USA), a heart rate monitor (HRM-Dual^®^, Garmin, Olathe, KS, USA), and an IMU on each limb (as per laboratory description) to capture field-based measurements. Participants were instructed to run at the same speeds as their laboratory treadmill run (preferred running speed), with audible feedback provided using GPS watches (if pace was ±5% of preferred running pace). Step rate and heart rate were recorded using the GPS watch. Axial PTA was collected onboard the IMU sensors using both the low-g (±16 g) and high-g (±200 g) accelerometers at 1125 Hz and 1600 Hz, respectively.

#### 2.4.3. Feasibility Outcomes

Feasibility was assessed through program retention, session adherence, and reported difficulty in achieving the target axial PTA thresholds. Program retention rates were evaluated as the percentage of completions at follow-up and retention testing sessions. Session adherence rates were assessed as the percentage of retraining sessions completed by analysing GPS watch records and participant training logs. Subjective data were recorded at the end of each retraining run to determine the perceived self-reported difficulty of achieving the target axial PTA threshold using a 100 mm visual analogue scale (VAS) with anchor points consisting of “no difficulty” on the left and “unable to perform” on the right end [45].

#### 2.4.4. Adverse Events

Adverse events included documenting any self-reported pain and injury. Participants recorded pain experienced throughout each training run using 100 mm VAS scales for seven regions of interest with anchor points consisting of ‘no pain’ on the left and severe pain’ on the right end [46]. Regions of interest included the foot, ankle, calf, shin, knee, thigh, and lumbopelvic area. Occurrences of running-related injuries were evaluated using the participants’ training log and through regular contact with participants. A running-related injury was defined as a new pain or discomfort that caused a restriction on or stopped running for at least seven days, for three consecutive scheduled training sessions, or requiring the runner to consult a physician or other health professional [47]. Participants were withdrawn if they reported any running-related injuries throughout the study.

### 2.5. Data Processing

Field data were collected using the IMU sensors (axial PTA_Field_) and Garmin GPS watch (heart rate and step rate). Axial PTA_Field_ was analysed using the low-g sensor due to its higher fidelity capabilities and reduced noise compared to the high-g sensor [48]. If impacts exceeded 16 g, data were analysed using the high-g sensor. Acceleration data were obtained throughout the 2.7 km run and processed using MATLAB (R2022a, MathWorks, Inc., Natick, MA, USA). Data were filtered using a low-pass, fourth-order Butterworth recursive filter with a cut-off frequency of 75 Hz to ensure only non-physiological frequencies were removed from the accelerometry signal [28]. The mean axial PTA, heart rate, and step rate across the complete field run were reported. Axial PTA_Field_ was normalised to gravitational acceleration and reported as multiples of gravity (g).

Biomechanical data collected during laboratory testing were processed using Vicon NEXUS (Vicon, Oxford, UK). Custom algorithms were used to label markers, which were manually checked for accuracy. Gaps or missing marker trajectories (<12 frames) were filled using a Woltring quintic spline [49]. Data were exported to Visual 3D (v6; HAS-Motion, Kingston, Ontario, Canada), where a seven-segment lower extremity model was built to determine kinematic and kinetic variables. Calibration markers at the ankle and knee were used to define joint centres [50]. Hip joint centres were predicted using anterior and posterior iliac spines [51]. Knee and ankle joint centres were predicted using the midpoint of the medial and lateral femoral condyles, and the malleoli at the knee and ankle, respectively. Kinematic and force plate data were filtered with a matched cut-off frequency of 20 Hz [52] via a fourth-order Butterworth low-pass filter [53]. Gait events were determined using a vertical ground reaction force (GRF) threshold of 50N [54]. Acceleration data from IMUs were filtered and reported using methods identical to those of the field data. IMU global angle data were not captured because doing so required the devices to reduce the accelerometer sampling rate below the required rate. Three-dimensional motion capture is the gold standard for running kinematic assessments and was used to assess ankle angles during our laboratory testing.

Kinetic variables included axial PTA_Lab_, vertical GRF, impact peak, average vertical loading rate (AVLR), and instantaneous vertical loading rate (IVLR) using the methods previously described [55]. The impact peak was defined as the first peak in the ground reaction force curve determined as the first point above 75% of a participant’s body weight with an instantaneous load rate of less than 15 body weights. IVLR was considered the maximum instantaneous slope, between 20% and 80% of the force at the impact peak. Kinematic variables included joint angles for ankle dorsiflexion (peak and initial contact), knee flexion (peak and initial contact), hip flexion (peak and initial contact), hip adduction (peak), and peak contralateral pelvic drop. Spatiotemporal variables included step length and step rate (steps per minute). All laboratory biomechanical variables were averaged over a 30 s interval during the final minute of running for each trial, recording at least 30 consecutive steps to ensure appropriate data stability [56]. Only data from the limb with the highest axial PTA during the screening was analysed across each time point.

Physiological and perceptual variables included oxygen consumption, heart rate, and RPE. Oxygen consumption and heart rate were analysed over one minute during the final minute of running and reported as mL/min/kg and bpm [57]. RPE (6–20) were recorded during the last minute of running.

### 2.6. Statistical Analysis

As a feasibility study not powered a-priori to detect statistical significance, only descriptive analysis was undertaken. No *p*-values were reported for observed differences due to the high potential for type II error. Descriptive analyses were performed in Jamovi (v2.3.28) ([58] and Microsoft Excel (Microsoft Corporation, Redmond, WA, USA, 2018)). For each outcome, mean ± SD at baseline was described. Additionally, linear mixed models were used to describe the changes observed over time to determine the mean difference from baseline to follow-up and retention time points with 95% confidence intervals. Models included time as a fixed effect and participant as a random effect. Changes in kinematics and spatiotemporal outcomes were compared to respective minimal detectable change (MDC) values to explore relevance with respect to measurement error [59,60]. To explore practical relevance, changes in kinetic, clinical (pain and difficulty), and physiological outcomes were compared to minimal clinically important differences (MCID) from the literature. For kinetic outcomes, a difference of ≥15% was considered as the MCID, in accordance with Milner et al. [14]. For changes in VAS pain, an MCID of 13 mm was considered clinically relevant [61]. Due to a lack of data pertaining to the patient-reported difficulty scale, a change score of 13 mm was also considered clinically relevant. A change in physiological outcomes (oxygen consumption and heart rate) greater than 2.4% was considered the MCID [62]. For BORG RPE, an MCID of 1.5 was considered clinically relevant [63].

## 3. Results

### 3.1. Participant Demographics

The flow of participants through the study is illustrated in Figure 4. Eight participants completed baseline testing, but one withdrew and did not commence the intervention. Therefore, seven participants commenced the retraining program (four males and three females; mean ± SD age = 34.9 ± 11.3 years; height = 1.71 ± 0.11 m; body mass = 65.4 ± 14.1 kg; weekly distance = 38.3 ± 21.6 km; 5 km time = 23.3 ± 2.4 min; preferred running speed = 3.17 ± 0.20 m/s; screening PTA (highest limb) = 12.4 ± 1.6 g). Body mass was similar in magnitude between baseline and both follow-up (mean ± SD: 65.1 ± 14.2 kg) and retention (mean ± SD: 65.2 ± 15.0 kg) time points across participants.

### 3.2. Feasibility Outcomes and Adverse Events

All seven participants who commenced the gait retraining program completed all retraining sessions with 100% adherence to the axial PTA biofeedback method. The program was completed in an average ± SD duration of 22.1 ± 8.0 days. All participants completed the follow-up testing session post-retraining (completion rate: 100%). After returning to usual training following the completion of gait retraining, one participant (P2) reported a running-related injury (diagnosed by a physical therapist as plantar heel pain) and did not complete the retention testing at one month (completion rate: 85%).

The self-reported difficulty by participants in achieving their target axial PTA threshold reduced from a mean score of 56/100 mm during the first retraining session to 36/100 mm by the final session (Figure 5). Notably, one participant (P1) reported 80–100/100 mm difficulty with all sessions and identified challenges in consistently achieving the target axial PTA using any of the suggested retraining strategies.

Reported pain scores during runs were generally low, with mean scores less than 5/100 mm across all lower limb areas, apart from the calf region, where 4/7 (57%) participants reported pain. Calf pain peaked between sessions 4–7 with a mean score of 18/100 mm across runners who reported pain (range 10–40/100 mm) (Figure 6). Calf pain scores were reduced to a mean of 10/100 mm by the final session.

Runners rated their perceived difficulty in achieving their target axial PTA threshold during each retraining session on a scale of 0 to 100 (0 = no difficulty, 100 = unable to perform). The bold black line indicates the study mean, and the coloured lines represent individual participant responses.

Runners rated perceived pain in the calf during each retraining session on a Visual Analog Scale (VAS) of 0 to 100 (0 = no pain, 100 = worst pain imaginable). The bold black line indicates the study mean, and the coloured lines represent individual participant responses.

### 3.3. Laboratory Testing Outcomes

Laboratory outcomes for running biomechanical and economy measures are displayed in Table 1**.**

#### 3.3.1. Running Kinetics

Across all participants, the mean reduction in axial PTA_Lab_ between the baseline and follow-up was 3.1 g (−33.3%) and remained 3.2 g (−34.4%) lower than the baseline at retention (Figure 7). Decreases in Axial PTA_Lab_ were observed across all participants at follow-up, with 5/7 participants demonstrating clinically relevant reductions (a difference of ≥15% from baseline). At one month, 4/6 participants displayed clinically relevant reductions in axial PTA_Lab_, with two participants (P1 and P6) demonstrating an Axial PTA_Lab_ similar to the baseline values. Mean IVLR reduced by 37.1 BW/s (−36.3%) between baseline and follow-up, with reductions maintained at one month. All participants demonstrated reduced IVLR at follow-up, with clinically relevant reductions among 6/7 participants. At retention, across all participants, IVLR remained at least 14% lower than the baseline, with 4/6 achieving clinically relevant reductions. Similarly, AVLR reduced by 33.2 BW/s (−41.7%) at follow-up and 32.8 BW/s (−41.2%) at retention, with 6/7 participants achieving clinically relevant reductions at follow-up and all participants (6/6) at retention. No change was observed for impact peak and peak vertical GRF following retraining.

#### 3.3.2. Running Kinematics

Across all participants, ankle dorsiflexion angle at initial contact decreased (indicating more plantarflexion) by over 10° between the baseline and follow-up, with 6/7 participants adopting a meaningful change. At retention, 4/6 participants also maintained a meaningful decrease. Only small decreases of less than 3° from the baseline were observed for peak ankle dorsiflexion and peak knee flexion at follow-up, with neither decrease maintained at the retention testing. There were also small reductions in peak hip adduction angle (indicating less hip adduction) at follow-up and contralateral pelvic drop at retention, but with high variability between participants. No other notable kinematic changes were observed.

#### 3.3.3. Spatiotemporal Outcomes

Mean differences in step rate were similar in magnitude between the baseline and follow-up sessions across participants, with an increase of approximately five steps per minute, below the clinically relevant level. At follow-up, two participants (P2 and P6) demonstrated a large clinically relevant increase in step rate of 9–14 steps per minute (+5–8% from the baseline), which remained 10 steps per minute higher than the baseline at retention (P6 data only). Mean step length did not considerably change between baseline and follow-up or retention time points (≤0.03 m).

#### 3.3.4. Running Economy Outcomes

Mean differences in oxygen consumption at follow-up demonstrated a clinically important change, with a 2.4 mL/min/kg increase from the baseline (+5.6%). Oxygen consumption decreased between follow-up and retention by 4.0 mL/min/kg (−8.7%). At retention, there was a clinically important improvement in oxygen consumption from baseline (−3.7%) across all participants. However, individual results varied, with 3/6 runners demonstrating clinically important reductions in oxygen consumption from the baseline (≤7%), indicating improved running economy, while 2/6 participants had increases (4–5%). There were no clinically relevant changes in heart rate or RPE following retraining.

### 3.4. Field Testing Outcomes

Field outcomes are displayed in Table 2. Mean axial PTA_Field_ reduced by 3.2 g at follow-up (−29.4%) and remained 2.6 g lower than the baseline at the retention testing (−23.9%). At follow-up, axial PTA_Field_ was lower than baseline levels across all participants, with 4/7 displaying reductions of at least 20% of baseline. At one month, axial PTA_Field_ remained at least 20% lower than the baseline in four participants, while one participant (P6) reverted to baseline levels, and an additional participant (P3) exceeded their baseline levels by 17%. Mean step rate increased from the baseline to follow-up by 4.3 steps per minute (+2.5%) and remained 5.9 steps per minute (+3.4%) higher than the baseline at retention. No meaningful changes were observed in running speed or heart rate across testing sessions.

IVLR (A), Axial PTA_Lab_ (B), and Axial PTA_Field_ (C) changes between baseline, follow-up, and retention testing. The bold black line indicates study mean, and coloured lines represent individual participant responses. Abbreviations: g, gravity; IVLR, instantaneous vertical loading rate; PTA_Field_, field peak tibial acceleration; PTA_Lab_, laboratory peak tibial acceleration.

## 4. Discussion

This study investigated the feasibility of field-based gait retraining with tibial-mounted accelerometers to reduce impact loading in recreational runners. Our observations support the feasibility of an eight-session field-based gait retraining program using axial PTA. High program retention rates and adherence to retraining sessions within the study are evidence of feasibility. However, with over 50% of participants reporting some level of calf pain and one participant withdrawing due to a foot injury, future studies using PTA-based gait retraining should be implemented with careful consideration of appropriate recovery periods between sessions to mitigate the potential risk of injury or discomfort experienced by runners.

Following the field-based retraining program, we found that runners typically decreased their axial PTA and vertical loading rates. Overall, these reductions in axial PTA were evident across both field and laboratory settings and were maintained at one-month follow-up after runners returned to their usual training. In laboratory testing, runners adopted a more forefoot landing pattern than the baseline to reduce their axial PTA, with kinematic changes maintained at one-month follow-up. Finally, there was a trend towards increased oxygen consumption following gait retraining, which returned to baseline levels by one month.

### 4.1. Kinetic Changes Following Gait Retraining

We found that mean axial PTA decreased from the baseline during both field (29.4%) and laboratory (33.3%) sessions following retraining. These reductions in axial PTA were maintained at one-month post-retraining, remaining 23.9% lower than the baseline in field settings and 34.4% lower than the baseline in laboratory settings. Our results were consistent with previous studies that provided feedback on axial PTA in laboratory settings and reported reductions between 26 and 32% [29,31,33]. However, in our study, individual responses varied, with one runner (P3) demonstrating higher PTA_Field_ than the baseline at one month despite showing reductions in PTA_Field_ at follow-up. This indicates that in some participants, further retraining sessions may be required in order to maintain reductions in axial PTA in field settings. Our findings suggest that the retraining program is feasible, and its effectiveness should be explored in future RCTs

We observed a reduction in vertical loading rates in laboratory testing accompanying the reductions observed for axial PTA following retraining. Many previous studies required a laboratory to implement gait retraining interventions or to provide real-time feedback on vertical loading rates. Given the close association between vertical loading rates and axial PTA [27], it is likely that vertical loading rates in the field will also be reduced. This suggests that field-based retraining using axial PTA feedback can be used as an alternative to laboratory-based retraining, which can improve accessibility for runners and can be performed in their own running environments. Additionally, using field-based retraining programs can facilitate large trials without requiring access to sophisticated laboratory equipment. This is particularly promising given the reduced injury rates previously reported by RCTs that investigated laboratory-based loading rate retraining.

### 4.2. Kinematic and Spatiotemporal Changes Following Gait Retraining

Several kinematic and spatiotemporal changes accompanied reductions to axial PTA and vertical loading rates, with runners decreasing ankle dorsiflexion on initial contact (indicating a more forefoot strike) and a small decrease in peak ankle dorsiflexion and knee flexion. Our findings were consistent with previous laboratory [41] and overground [31] retraining studies that instructed runners to reduce their axial PTA and identified that runners achieved this by altering their ankle joint mechanics and footstrike. Lower axial PTA values have been observed in runners with a forefoot strike compared to a rearfoot strike [64,65]. Transitioning to a forefoot strike has also been used as a strategy to reduce runners’ loading rates [66]. It is clear that the runners in our study identified footstrike ankle changes as an effective method they could use to reduce axial PTA. These changes were maintained in laboratory settings one-month post-retraining, suggesting that this kinematic strategy may be retained. Additionally, runners in our study adopted a small increase in step rate following gait retraining, though not at a clinically important level. Numerous studies have previously instructed runners to increase their step rate as a gait retraining strategy, typically by 7.5–10% [67,68,69], and observed reductions in impact-based variables following retraining. However, the changes we observed in step rate were relatively small, between 2–3% (~5 steps per minute), indicating that step rate changes may not be the primary driver of the impact-related changes observed in our study.

In our feasibility study, participants were not instructed to modify a single discrete variable to reduce their axial PTA, which is common in previous retraining studies (e.g., increased step rate, transition to forefoot footstrike). Instead, they were allowed to self-optimise their own solution to achieve a reduction in axial PTA. Apart from a single participant (P1), all runners perceived difficulty in achieving their target axial PTA threshold reduced following retraining. This suggests that the design of the retraining program is appropriate for a future trial. However, restricting all runners to the same program may reduce the ability to achieve the desired gait changes in a small portion of runners who find gait adjustments challenging. Future trials addressing this should consider a semi-standardised retraining program that allows some scope for individual tailoring.

### 4.3. Running Economy Outcomes Following Gait Retraining

Following gait retraining, a clinically meaningful 5.6% increase in mean oxygen consumption was observed, indicating reduced running economy. However, oxygen consumption returned below baseline levels by the retention session, indicating no prolonged detriments and, in fact, potential improvements to running economy. This finding suggests a less economical running pattern directly following gait retraining, which is consistent with previous laboratory findings that found acute alterations of running technique can lead to an increase in metabolic cost [35] and cognitive demand [34]. Participants also initially reported difficulty with achieving their target axial PTA threshold. Perceived difficulty reduced as retraining sessions progressed and feedback faded, indicating that participants may have become accustomed to the new gait pattern. Interestingly, we observed an improvement in oxygen consumption between the follow-up session and the one-month retention session, which suggests that participants may have adapted to the retrained gait and re-established their previous level of running economy. This is important as runners prescribed gait retraining may become discouraged if the perceived effort is too great or if runners experience reductions in running economy and performance. A target axial PTA threshold of 20% reduction from baseline was achievable for most runners but may not be attainable by all. Further trials should consider a graded reduction approach if the perceived difficulty is too challenging to ensure the program does not demotivate runners. Additionally, trials should consider providing educational programs to inform runners that they may experience short-term reductions in running economy following gait retraining, but that these changes are not expected to be prolonged.

### 4.4. Adverse Events Following Gait Retraining

Over half of the runners in the study reported a small increase in calf pain during the gait retraining program, which was reduced in magnitude by the final retraining session. One participant (P2) who reported calf pain went on to experience a foot injury following a return to usual training and withdrew. This participant reported a rapid increase in running volume after completing the gait retraining program. Subjective reports of calf soreness were identified in previous laboratory gait retraining studies that attempted to reduce axial PTA [31] or instruct a forefoot strike pattern [70]. However, these studies did not quantify the level of soreness, making direct comparisons challenging [70]. The kinematic strategies to reduce axial PTA adopted by runners in our study included decreased ankle dorsiflexion on impact, resulting in a more pronounced forefoot strike than the baseline, which may place increased loading on the foot and calf complex. In our study, participants were advised to allow additional recovery days between the field-based retraining sessions to allow appropriate recovery time and reduce any reported soreness.

### 4.5. Future Applications

This feasibility study informs a vertical acceleration-based IMU method for translating lab-based injury prevention strategies to field settings. Our findings demonstrate that the efficacy of impact-based retraining should be assessed in larger studies using RCT design. Researchers and clinicians should be cautious when using impact-based gait retraining, as the kinematic strategies that reduce axial PTA and vertical loading rates may place increased load on the distal kinetic chain, such as the foot and calf complex [71]. Prescribed retraining programs should use a target axial PTA threshold of no more than 20% reduction from baseline to ensure it is achievable for runners. Careful monitoring for calf and lower limb pain and incorporating gradual increases in prescribed retraining loads are required to help mitigate impact-based retraining effects. Additionally, programs should incorporate faded feedback and ensure they allow runners sufficient recovery time between sessions and adequate time to adapt to the modified gait.

Future research should also consider incorporating a broader range of IMU features to maximise the potential of IMU technology in sensor-based gait analysis. While many studies have used accelerometers and other sensors to monitor and provide feedback on parameters like cadence and vertical loading, numerous other metrics remain unexplored [72]. Comprehensive analyses of these features could deepen our understanding of gait mechanics and enhance the effectiveness of gait retraining interventions.

### 4.6. Study Limitations

It is important to note that this study has some limitations. Firstly, it was a feasibility study with a relatively small sample size and did not have a control group. The small sample size means there is uncertainty about how our observation will generalise to a larger cohort of runners. Despite the small sample size, participant responses support future larger studies to explore the benefits further. Future studies should include a control group to confirm that the changes observed were related to the gait retraining and not simply a result of the structured running program.

Secondly, axial PTA indirectly estimates vertical loading rates based on laboratory studies and is not directly correlated with injury incidence. The effects of reducing axial PTA on injury prevention or management are currently unknown and should be explored further in large-scale prospective studies. Additionally, axial PTA was recorded during flat treadmill running or mostly flat outdoor running. Axial PTA is affected by incline or decline running [73] and will differ during outdoor running environments that involve hill ascents or descents.

## 5. Conclusions

We found that providing field-based axial PTA feedback is a feasible way to reduce axial PTA and vertical loading rates in high-impact recreational runners. This program did not adversely affect running economy. The reductions in axial PTA were evident across laboratory and field settings and were maintained at one month in all but one participant. Field-based gait retraining using axial PTA warrants further investigation in a large RCT. Our findings suggest that future trials using impact-based retraining can achieve high feasibility by using a target axial PTA reduction of no more than 20% from the baseline and should incorporate at least eight sessions with faded feedback, allow participants to self-optimise their retraining approach, monitor for calf and lower limb pain, and ensure runners receive sufficient recovery time to adapt to the retrained gait.

## Figures and Tables

**Figure 1 sensors-25-01712-f001:**
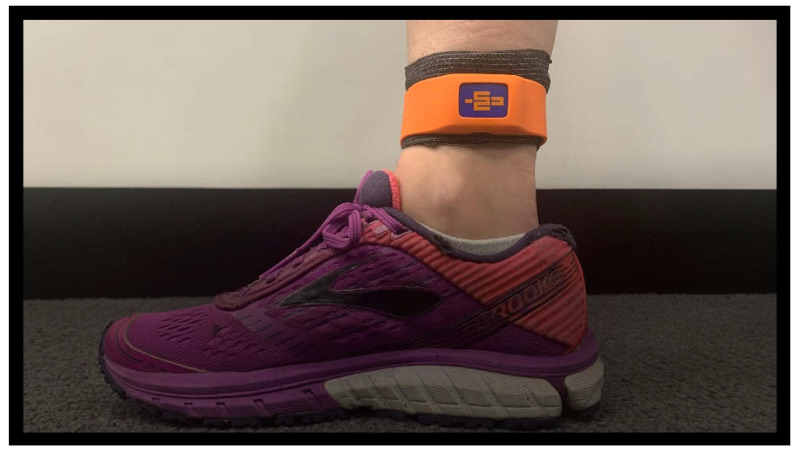
Inertial measurement unit (IMU) attachment on the right lower limb. Elastic cohesive taping was used to secure the IMU.

**Figure 2 sensors-25-01712-f002:**
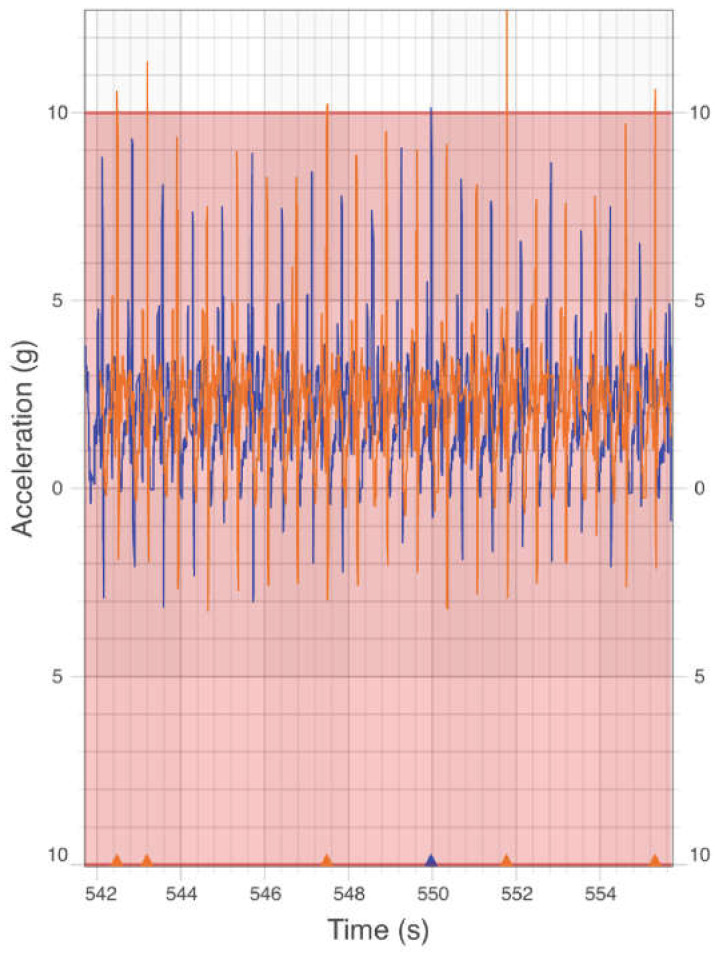
Real-time axial peak tibial acceleration (PTA) feedback. The blue lines indicate the left axial PTA, while the orange lines indicate the right axial PTA. In this example, the runner occasionally exceeds their axial PTA threshold of 10 g with the right limb.

**Figure 3 sensors-25-01712-f003:**
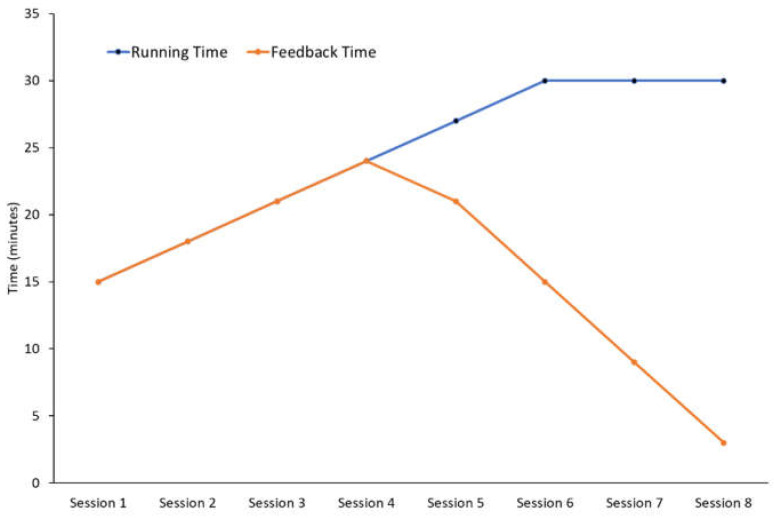
Schedule of running and feedback time over the eight field-based retraining sessions.

**Figure 4 sensors-25-01712-f004:**
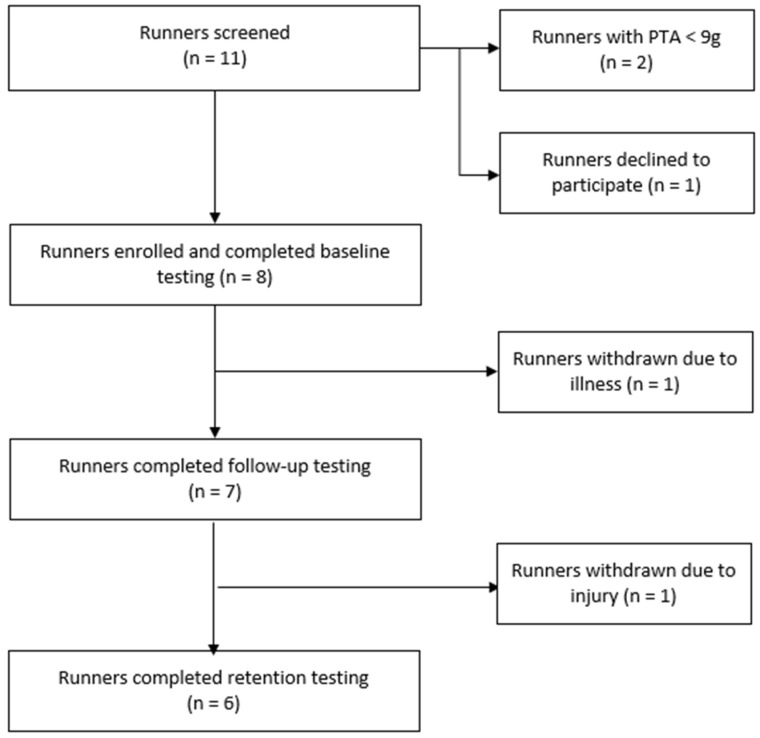
Flow chart of participants through the study. Abbreviations: PTA, peak tibial acceleration.

**Figure 5 sensors-25-01712-f005:**
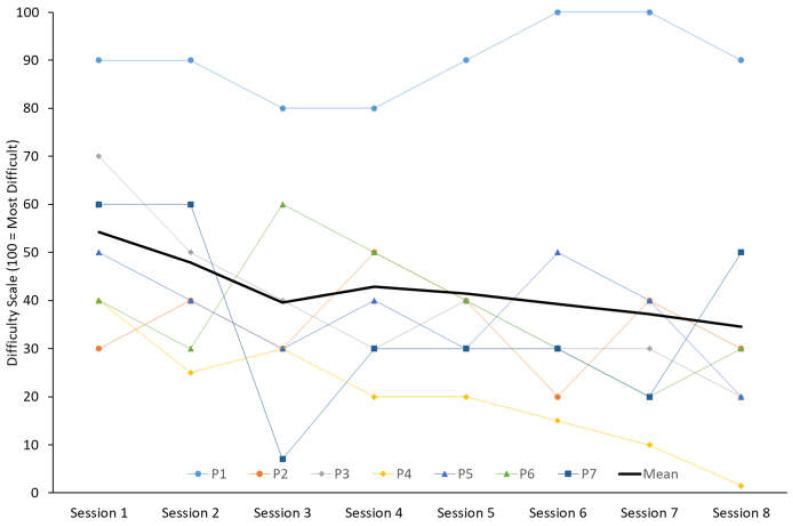
Participant-reported difficulty scale during retraining.

**Figure 6 sensors-25-01712-f006:**
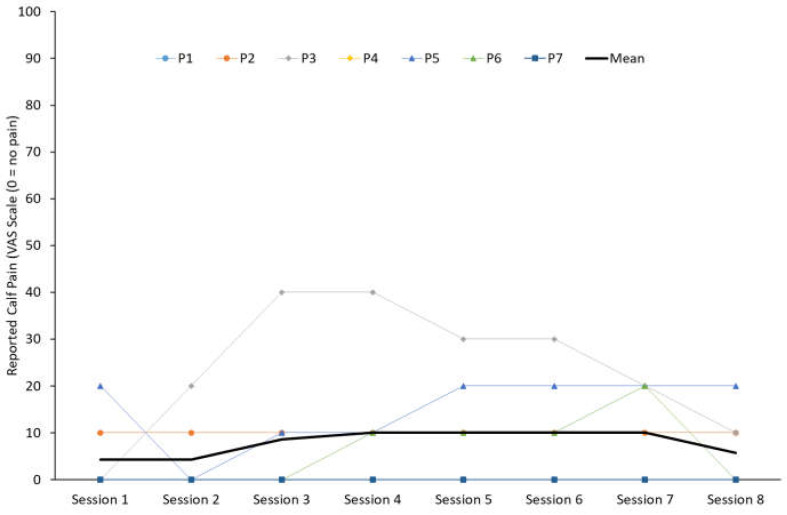
Participant-reported calf pain during each retraining session.

**Figure 7 sensors-25-01712-f007:**
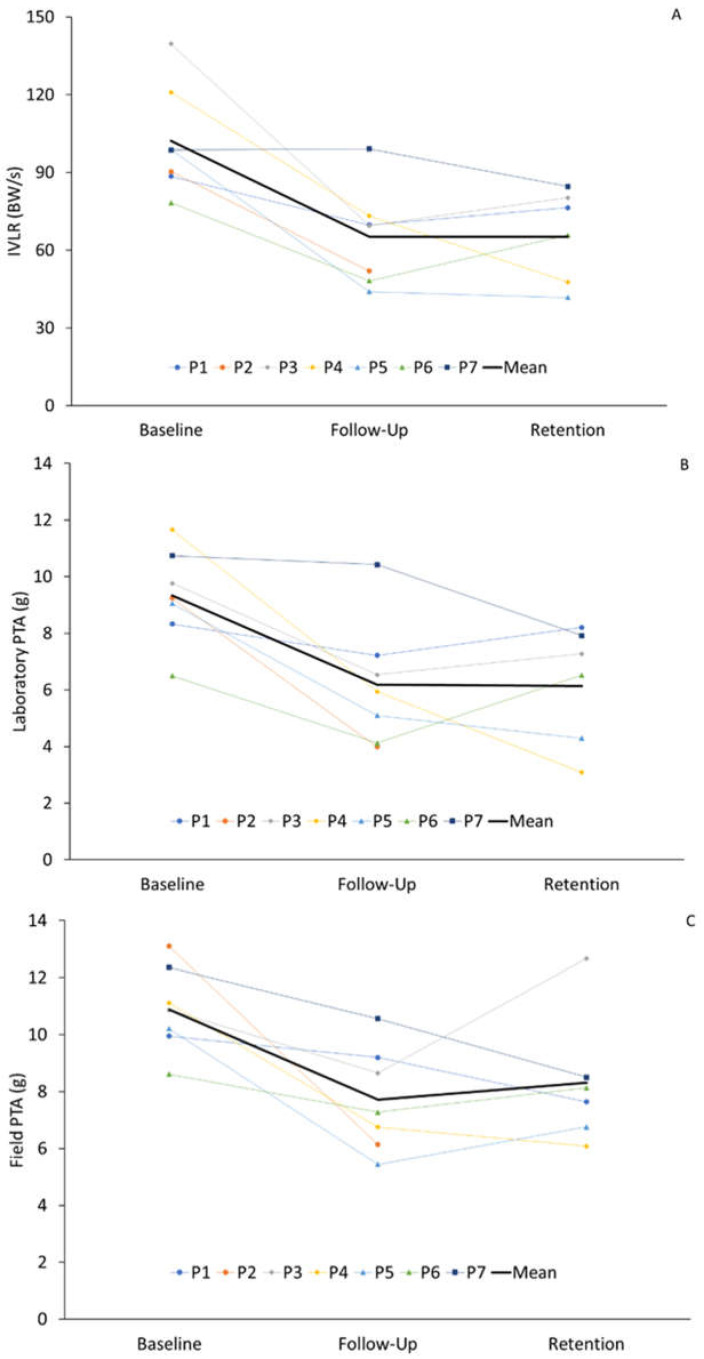
Axial PTA and IVLR changes following gait retraining.

**Table 1 sensors-25-01712-t001:** Mean (SD) scores for laboratory outcomes at each time point with mean difference changes over time (95% CI).

Variable	Baseline	MDC/ MCID	Follow-Up	Retention
	Mean (SD)		Difference from Baseline (95% CI)	Difference from Baseline (95% CI)
**Kinetics**				
Axial PTA_Lab_ (g)	9.3 (1.7)	1.4 ^†^	−3.1 (−5.0 to −1.3) *	−3.2 (−5.1 to −1.3) *
AVLR (BW/s)	79.7 (19.3)	12.0 ^†^	−33.2 (−49.3 to −17.0) *	−32.8 (−49.7 to −15.9) *
IVLR (BW/s)	102.2 (21.1)	15.3 ^†^	−37.1 (−54.0 to −20.2) *	−37.1 (−54.8 to −19.3) *
Impact peak (BW)	1.82 (0.26)	0.27 ^†^	−0.02 (−0.32 to 0.29)	0.11 (−0.21 to 0.44)
Peak vertical GRF (BW)	2.54 (0.23)	0.38 ^†^	0.05 (−0.01 to 0.11)	−0.03 (−0.09 to 0.04)
**Kinematics**				
Ankle dorsiflexion—initial contact (°)	−2.4 (6.4)	6.9 ^‡^	−11.1 (−15.5 to −6.7) *	−8.1 (−12.8 to −3.5) *
Ankle dorsiflexion—peak (°)	17.8 (4.2)	3.1 ^‡^	−3.0 (−4.3 to −1.6)	−2.0 (−3.4 to −0.5)
Knee flexion—initial contact (°)	14.4 (2.9)	6.9 ^‡^	−1.3 (−2.6 to 0.0)	−0.9 (−2.3 to 0.4)
Knee flexion—peak (°)	37.8 (4.6)	5.3 ^‡^	−2.5 (−4.2 to −0.9)	−1.8 (−3.5 to 0.0)
Hip flexion—initial contact (°)	34.2 (6.5)	N/A	−0.5 (−3.0 to 2.1)	1.5 (−1.2 to 4.3)
Hip flexion—peak (°)	36.2 (7.2)	5.6 ^‡^	−1.7 (−4.6 to 1.2)	0.0 (−3.0 to 3.1)
Hip adduction—peak (°)	10.5 (3.2)	1.8 ^‡^	−3.0 (−5.1 to −0.9)	−1.2 (−3.4 to 1.0)
Contralateral pelvic drop (°)	10.9 (5.8)	1.7 ^‡^	−1.2 (−3.8 to 1.4)	−1.8 (−4.5 to 1.0) *
**Spatiotemporal**				
Steps rate (steps/min)	175.5 (10.9)	5.7 ^‡^	4.9 (2.0 to 7.8)	4.8 (1.7 to 7.9)
Step length (m)	1.09 (0.10)	0.10 ^‡^	−0.03 (−0.04 to −0.01)	−0.03 (−0.04 to −0.01)
**Running Economy**				
Oxygen consumption (ml/min/kg) ^	43.2 (6.8)	1.0 ^§^	2.4 (0.02 to 4.8) *	−1.6 (−4.1 to 1.0) *
Heart rate (bpm)	154.7 (11.1)	3.7 ^§^	−3.6 (−6.8 to −0.4)	−1.2 (−4.6 to 2.2)
RPE (Borg 6–20)	11.9 (0.9)	1.5 ^§^	0.0 (−0.4 to 0.4)	0.1 (−0.3 to 0.6)

* Mean difference exceeds respective MDC or MCID; ^†^ Denotes a MDIC of ≥15% for axial PTA [14] and 2.4% for heart rate [62]; ^‡^ Denotes a MDC based on healthy runners [60]; ^ Steady-state oxygen consumption at preferred running speed; ^§^ = Oxygen consumption. Abbreviations: bpm, beats per minute; CI, confidence interval, m, meters; MCID; minimum clinically important difference; MDC; minimum detectable change; N/A, not applicable; PTA, peak tibial acceleration; s, seconds; spm, steps per minute.

**Table 2 sensors-25-01712-t002:** Mean (SD) scores for field outcomes at each time point with mean difference changes over time (95% CI).

Variable	Baseline	MDC/ MCID	Follow-Up	Retention
	Mean (SD)		Difference from Baseline (95% CI)	Difference from Baseline (95% CI)
Running speed (m/s)	3.2 (0.3)	N/A	−0.1 (−0.5 to 0.2)	0.1 (−0.3 to 0.5)
Axial PTA_Field_ (g)	10.9 (1.5)	1.6 ^†^	−3.2 (−4.9 to −1.41) *	−2.6 (−4.4 to −0.7) *
Step rate (steps/min)	173.7 (11.2)	5.7 ^‡^	4.3 (1.7 to 6.8)	5.9 (3.1 to 8.6) *
Heart rate (bpm)	160 (12)	3.8 ^†^	−3 (−9 to 2)	1 (−5 to 7)

* Mean difference exceeds respective MDC or MCID; ^†^ Denotes a MDIC of ≥15% for axial PTA [14] and 2.4% for heart rate [62]; ^‡^ Denotes a MDC based on healthy runners [60]. Abbreviations: bpm, beats per minute; CI, confidence interval; g, unit of gravity; m, meters; min, minute; MCID, minimum clinically important difference; MDC, minimum detectable change; N/A, not applicable; PTA, peak tibial acceleration; s, seconds.

## Data Availability

Data are available on request from the authors. The data supporting this study’s findings are available from the corresponding author upon reasonable request by emailing Eoin Doyle (eoin.doyle@mq.edu.au).

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
