# Peer review of "Field-Based Gait Retraining to Reduce Impact Loading Using Tibial Accelerometers in High-Impact Recreational Runners: A Feasibility Study"

_sensors, 2025, doi:10.3390/s25061712_

Round 1
Reviewer 1 Report
Comments and Suggestions for Authors
This paper provides valuable insights into the feasibility and effectiveness of gait retraining using tibial-mounted accelerometers. Although the analyses were done thoroughly, the number of effective participants was still low, making the statistical results not solid enough. Please comment on this issue.
Reviewer 2 Report
Comments and Suggestions for Authors
This article investigates field-based gait retraining to reduce impact loading using tibial accelerometers in high-impact recreational runners. The structure of the article is commendable, featuring thorough analysis and a well-executed experimental study. The paper is well-organized, facilitating easy understanding. Several suggestions are provided below to enhance its quality further.
Firstly, if the approach described in this article were to be translated into a commercial product with corresponding algorithm software, it is essential to prioritize variable control in the research phase. Gait analysis is influenced by numerous factors, and controlling one variable at a time is crucial for obtaining accurate and reliable results. For instance, the study should maintain the PTA (Peak Tibial Acceleration) variable constant while analyzing its relationship with other variables, including individual factors such as body weight. Since PTA is directly measured using an Inertial Measurement Unit (IMU), the innovation of this study lies in utilizing IMU data to correct gait patterns. Additionally, the research should examine the implementation rate of PTA, address potential challenges, and explore how individual differences, such as varying body weights, influence the outcomes.
Secondly, the primary focus of this paper should be on using IMUs to correct gait, thereby replacing traditional sensor methods. Currently, the study relies solely on the peak values obtained from IMU data to analyze gait, while other gait characteristics depend on laboratory-based software. This approach limits the full potential of IMU technology. As a publication in a sensor-focused journal, the emphasis should be placed on comprehensive sensor data analysis. IMU data can be leveraged not only to assess kinematic parameters (running kinetics) but also to analyze various aspects of movement. For example, waveform features from IMU data can be used to infer ankle dorsiflexion angles during running. Incorporating a broader range of IMU feature analyses would significantly enhance the study's contribution to sensor-based gait analysis.
Overall, the paper is of high quality, presenting valuable insights into gait retraining using tibial accelerometers. To further strengthen the study, more attention should be given to leveraging comprehensive IMU data for gait retraining. Expanding the feature analysis beyond peak values will provide a more detailed understanding of gait mechanics and enhance the effectiveness of the proposed retraining interventions.
Reviewer 3 Report
Comments and Suggestions for Authors
This paper explores the feasibility of using real-time axial peak tibial acceleration (PTA) feedback for an on-site gait retraining program in high-impact recreational runners and analyzes its impact on running biomechanics and economics. Moreover, it seems that the desired effect is realized from the results. However, there are still some language and formatting problems in the paper. Thus, a major revision is required, and the final decision cannot be made until the following problems are solved.
The following are the comments.
- The innovation of this paper is not clearly highlighted. It is recommended that the authors clearly summarize the key innovations of the paper and present them in bullet points in the introduction section for better clarity.
- Figures 1 and 2 in this paper lack visual appeal. It is recommended that the authors optimize these figures to enhance their overall aesthetics and improve the presentation.
- Some subsection headings in this paper are too short, consisting of only one word. It is recommended that the author revise these headings to provide more context.
- Some tables in this paper extend beyond the page margins. It is recommended that the authors adjust these tables to ensure they fit within the margins.
- In this paper, the change in the P3 curve in Figure 7(C) differs significantly from the others. The authors are asked to provide a more detailed analysis of the reasons behind this discrepancy.
Round 2
Reviewer 3 Report
Comments and Suggestions for Authors
I have no further comment. This paper can be accepted.
Comments on the Quality of English LanguagePlease polish the language issues. I have no further comment. This paper can be accepted.